# Mirror-gazing-induced dissociation impairs self-reported and implicit sense of agency: A causal investigation of dissociation and agency under controlled laboratory conditions

Noa Bregman-Hai ⓘ *, Nirit Soffer-Dudek ⓘ

Department of Psychology, Ben-Gurion University of the Negev, Beer-Sheva, Israel

* noabreg@post.bgu.ac.il

## Abstract

Dissociation and sense of agency disruptions are related, but the causal pattern between them is unknown. This two-phase study explored whether laboratory-induced dissociative experiences impair the sense of agency. Study 1 tested the feasibility of generating dissociation and investigated its effect on explicit (self-reported) agency. $N = 98$ undergraduate students underwent one of two mirror-gazing manipulation conditions (with or without a suggestive cue) or a control condition (video watching). Mirror-gazing induced dissociation and decreased agency, especially among high trait dissociators, but the control condition was not sufficiently differentiated, as it also increased absorption, a construct which is considered a part of the domain of dissociative experiences. In Study 2, $N = 199$ undergraduates underwent the same manipulations with an additional improved control condition (article reading) and performed a task assessing agency implicitly. Mirror-gazing impaired explicit agency only among those high in self-reported trait dissociation, whereas it impaired implicit agency generally, perhaps bypassing self-report bias. Reading proved to be a better control condition for dissociation induction. The findings provide pioneering evidence of the causal relationship between dissociation and impairments of the sense of agency. Clinical implications are discussed.

Dissociation is a psychological phenomenon characterized by a lack of integration between mental processes which are usually integrated, such as consciousness, memory, identity, emotion, perception, body representation, motor control, or behavior [1]. Dissociative symptoms are associated with various dysfunctions in everyday life [2,3], but the dynamics causing those dysfunctions are poorly understood. One of the pathways through which dissociation may affect daily functioning is through its effect on the sense of agency, which is the feeling of control over our actions, their

**Data availability statement:** Anonymized data are openly available in Open Science Framework at https://doi.org/10.17605/OSF.IO/KU2MX.

**Funding:** The author(s) received no specific funding for this work.

**Competing interests:** The authors have declared that no competing interests exist.

initiation, and their consequences [4]. A solid sense of agency is important for performance and well-being [5]. Dissociative experiences and disruptions in the sense of agency both pertain to a distorted, fragmented experience of reality and of the self. Decreased sense of agency is even mentioned as a possible criterion for Dissociative Identity Disorder and Depersonalization/Derealization Disorder [1]. Yet, the research investigating their relationship is still in its infancy. Recent literature reported empirical [6–9] and theoretical [10] associations between the phenomena, and suggested a shared cognitive mechanism [11]. However, no experimental studies have explored whether dissociation directly affects the sense of agency. The present study aimed to examine whether an increase in state dissociation would cause a decrease in the sense of agency, and whether this effect would be evident both for explicit (self-reported) and implicit measures of agency.

Due to its multi-faceted nature, dissociation is considered both a trait and a state. The trait and state levels are correlated [12,13] yet distinct, referring, respectively, to a stable tendency to experience dissociative symptoms versus transient dissociative experiences that vary temporally and are influenced by external and internal stimuli [14]. Trait dissociation is often assessed via three subtypes of experiences: dissociative amnesia (difficulty in recalling autobiographical information), depersonalization-derealization (DPDR; experiencing feelings of detachment from oneself or the world), and absorption and imaginative involvement (AII; a tendency for absolute immersion in a stimulus with neglect of the surrounding environment) [15]. The assessment of state dissociation usually focuses on DPDR [16], but see [13] and [17] for state AII assessments. The sense of agency is also defined both as a general tendency and as a temporal experience. The trait tendency is assessed via self-report [18]. The state level can be assessed explicitly or implicitly. Explicit assessment relies on asking people to rate how much their actions on a certain task relate to the outcomes [19] and thus might be biased by their expectations. The implicit assessment, which is less susceptible to bias, is inferred from certain features of behavior, such as the tendency to perceive voluntary actions as closer to their outcomes than they really are, an effect called "intentional binding" [20]. The intentional binding task is the most widely used measure of implicit agency [4] and was also used in the current investigation; thus, it will be presented elaborately later in this article. Another implicit assessment method, called "sensory attenuation", relies on the finding that outcomes of self-generated actions are perceived as less intense than outcomes of actions performed by others [21,22]. In sensory attenuation paradigms, participants are asked to rate the intensity of sensory input, and their level of agency is inversely inferred from these intensity ratings [23]. It was recently examined in the context of borderline personality disorder, which is associated with frequent dissociative symptoms [24].

The correlation between dissociation and the sense of agency was recently demonstrated in several naturalistic studies [7–9], but to determine causality, an experimental investigation inducing dissociative symptoms is needed. Among the methods that have been used to experimentally induce dissociation are dot-staring [25], mirror-gazing [25–27], audio-photic stimulation [28], and stimulus deprivation [28], with mirror-gazing and audio-photic stimulation being most effective [25,28],

especially among participants with high rates of trait dissociation. In the present study, we followed a previously published mirror-gazing protocol [26,27,29] to induce state dissociation. For a detailed review of the hypothesized mechanisms that may underlie the mirror-gazing paradigm, see S1 Text. Notably, while our study was inspired by Caputo's [26,27] work, we did not replicate his procedure identically. We made slight adjustments to accommodate our laboratory conditions and the conceptual framework of the current study, as our goal was to induce dissociation and examine its effects on the sense of agency, rather than to study the mirror-gazing paradigm itself. Indeed, Caputo [27] noted that the exact details of the setting, such as the level of room illumination, are not critical. The specific methodological similarities and differences are detailed in the Method section. We will now turn to describe a specific difference which we consider more salient.

At the time of designing this study, we noticed that in the original procedure [26] participants were informed in advance that they might notice perceptual changes in their reflections. We speculated that such a warning might have influenced Caputo's results as it contained a potentially suggestive cue. Participants, especially those who may be more easily influenced, may have reported perceptual alterations due to experimental demands. Elevated responsiveness to verbal suggestions, i.e., suggestibility, is associated with dissociative symptoms [30], and demand characteristics were found to facilitate anomalous experiences in mirror-gazing paradigms [31]. It is worth noting that Caputo addressed this issue in a recently published article [32]. In that study, he intentionally did not instruct participants to observe facial changes, and found that they still reported anomalous self-experiences following the mirror-gazing paradigm. However, that article was published after data collection for both phases of the current study had been completed. Moreover, we believe that the present study contributes to existing knowledge, as it is the first to compare, within the same research design and sample, participants receiving instructions that include a suggestive aspect with those receiving more general instructions. The two conditions were identical except for the suggestive component, which allowed us to isolate the effect of suggestion and examine its unique contribution. More specifically, our experimental conditions were (1) a "pure" mirror-gazing condition in which participants were given only a general disclaimer about a possible change in sensations in their forthcoming experience, and (2) a "fortified" condition, which also included a subtle suggestion regarding face perception, similar to the one described by Caputo [26].

Notably, a recent study examined the relationship between dissociation and the sense of agency with a similar induction method [33]. In that study, participants reported reduced feelings of agency over their own reflection. However, the manipulation and measurement merged dissociation with disruptions of agency, preventing the conclusion of a causal relation, and the design included only self-reported measures, which could be biased.

To conclude, we aimed to determine to what extent dissociative experiences impair the tendency to experience oneself as the initiator of one's actions and as responsible for the results of these actions. We attempted to induce dissociation in a controlled environment, using a mirror-gazing paradigm, and examine its direct effect on state levels of the sense of agency, assessed both explicitly, with self-report questions, and implicitly, with an intentional binding task. We also explored how the trait tendency for dissociation may moderate the causal relationship. We hypothesized that: (1) like previous studies, participants in the mirror-gazing condition, but not in the control condition, will experience an increase in state dissociation. (2) This will also lead to a decrease in state sense of agency. (3) A tendency to regularly dissociate will intensify these effects.

## Study 1 method

Study 1 was conducted as a pre-test, to test whether we could induce dissociation by adapting a version of Caputo's [26,27] mirror-gazing paradigm into our laboratory, and to examine the suitability of the control condition. We also wished to examine in a preliminary manner the central question of interest, namely, whether manipulating dissociation would affect the sense of agency. In this preliminary investigation, sense of agency was examined only explicitly through self-report. We also aimed to use this pre-test to explore the temporal persistence of the effects, to estimate the feasibility of performing the computerized task assessing implicit agency in Study 2 while remaining under the influence of the manipulation.

## Participants

Undergraduate Psychology students enrolled through the university's experiment system in exchange for course credit. The recruitment took place between October 5th, 2020, and November 21st, 2021. The study was approved in advance by the institutional IRB according to the principles of the declaration of Helsinki. The recruitment ad stated that the experiment required mentally healthy participants. Wearing eyeglasses was an exclusion criterion. One-hundred and nine undergraduates participated but 11 were excluded due to technical reasons detailed in the procedure section below, leaving 98 participants (79 females, mean age 23.9 years, SD = 1.03). This sample comprised Israeli students, all either native Hebrew speakers or fluent in Hebrew, mostly of middle-class or upper-middle-class socio-economic backgrounds. We determined our sample size based on the central limit theorem [34], i.e., a minimum of 30 participants per subgroup.

## Measures

Self-report questionnaires assessed state-level dissociation (DPDR and AII), state sense of agency, and trait dissociation.

**Trait dissociation.** The revised Dissociative Experiences Scale (DES-II) [15] requires estimating the percentage of time (0–100%, with intervals of 10% on an 11-point scale) in which each of 28 dissociative phenomena are experienced, and a total score is the average of all items. The Hebrew version of the DES has good psychometric properties [35]. Cronbach's alpha in this study was α = .90.

**State DPDR.** The Clinician-Administered Dissociative States Scale (CADSS) [16] asks participants to what extent they currently experience 19 DPDR phenomena. The intensity is rated on a 5-point scale (0 = "not at all", 4 = "extremely"). Since participants completed it several times within a relatively short period of time, we sought to reduce burden and prevent burnout. Thus, two items less compatible with the purpose of the study (item 8: "Do people seem motionless, dead, or mechanical?" and item 12: "Does this interview seem to be taking much longer than you would have expected?") were omitted from the questionnaire. The remaining 17 items were reliable at T1, T2, and T3 with α = .87,.87, and.85, respectively.

**State AII.** We used a 7-item measure [36] that estimates the current intensity of several AII experiences on an 11-point scale (0% = it does not reflect my current sensation at all, 100% = it reflects my current sensation exactly). It includes items such as: "In the last few moments, I found myself staring into space, not thinking about anything, and not aware of the passage of time". Internal reliability was α = .78 (T1),.84 (T2), and.80 (T3).

**State sense of agency.** A modified version of the Sense of Agency Scale (SoAS) [18] assesses agency as a state [7]. The SoAS is a 13-item trait questionnaire which includes statements such as: "My actions just happen without my intention" with a 7-point Likert scale, ranging from "1" (totally disagree) to "7" (totally agree). The adapted version tests current sensations rather than general ones (i.e.,: "At this moment, my actions just happen without my intention"). Internal reliability was α = .79 (T1),.87 (T2), and.84 (T3).

**Cognitive task.** A low-difficulty cognitive task [37] was used after the manipulation to assess the persistence of induced dissociation. We aimed to introduce a simple task that would not affect the level of dissociation but would only prolong the participants' stay in the laboratory. A highly demanding task might have taken the participants out of the induced dissociative state. In this computerized task, a target stimulus was presented on one side of the screen and participants were required to quickly press a key that indicates its location. In some of the trials, a congruent arrow-shaped clue was presented before the target. Since the task was only used to extend the time the participants stayed in the laboratory, we did not evaluate task performance.

## Procedure and experimental design

Before arrival, participants were randomly allocated to one of three conditions: video-watching control (V), mirror-gazing (MG), and mirror-gazing with suggestion (MGS), based on the procedure described by Caputo [26,27]. N = 109 participants were recruited, but the first 11 participants of the MG condition were excluded from analyses since they were not

sitting at a fixed distance from the mirror. Thus, 98 participants were included in the analyses: 32 participants in the control condition, 30 in the MG condition, and 36 in the MGS condition.

Participants arrived at the laboratory individually and provided written consent in an online form, which was kept separately from their experimental data. A research assistant witnessed the consent process and informed participants that they may feel unusual sensations during the study. This was a general disclaimer given to all groups, with no specific information given about modification of face perception. It was clarified that such sensations are harmless and if appear, will last only a few minutes. Then, participants answered a battery of state questionnaires assessing their baseline levels of dissociation and sense of agency (T1 measurement) and underwent an experimental manipulation (V, MG, or MGS).

In all conditions, participants sat alone in a dimly lit room located within the laboratory, while the experimenter waited in the larger outer laboratory room. Illumination was provided by a desk lamp with an adjustable arm holding a 25 W incandescent bulb. The lamp was placed on the floor in a corner behind the participant, oriented downward to provide indirect lighting. Due to the configuration of our laboratory room, the lamp was intentionally positioned off-center to prevent its reflection in the mirror in front of the participant. In the MG and MGS conditions, a 51 x 48.5 cm mirror was placed 40 cm in front of the participants, at an angle that did not reflect the light source (See Fig 1, Side A). Similar to several other studies that used mirror-gazing [38–45], we did not measure the level of face illumination during the task. Following a comment from an anonymous reviewer, we measured this retrospectively, and found the light intensity to be in the range of 3–5 lux, which is somewhat higher than the range of 0.6–1 lux described by Caputo [26,29].

In the MG condition, participants were instructed to gaze at their faces in the mirror. In the MGS condition, participants received the same instructions, followed by the sentence: "When looking in the mirror, you might experience illusions related to the way you see your face". Notably, we asked participants to gaze at their faces rather than focus on their

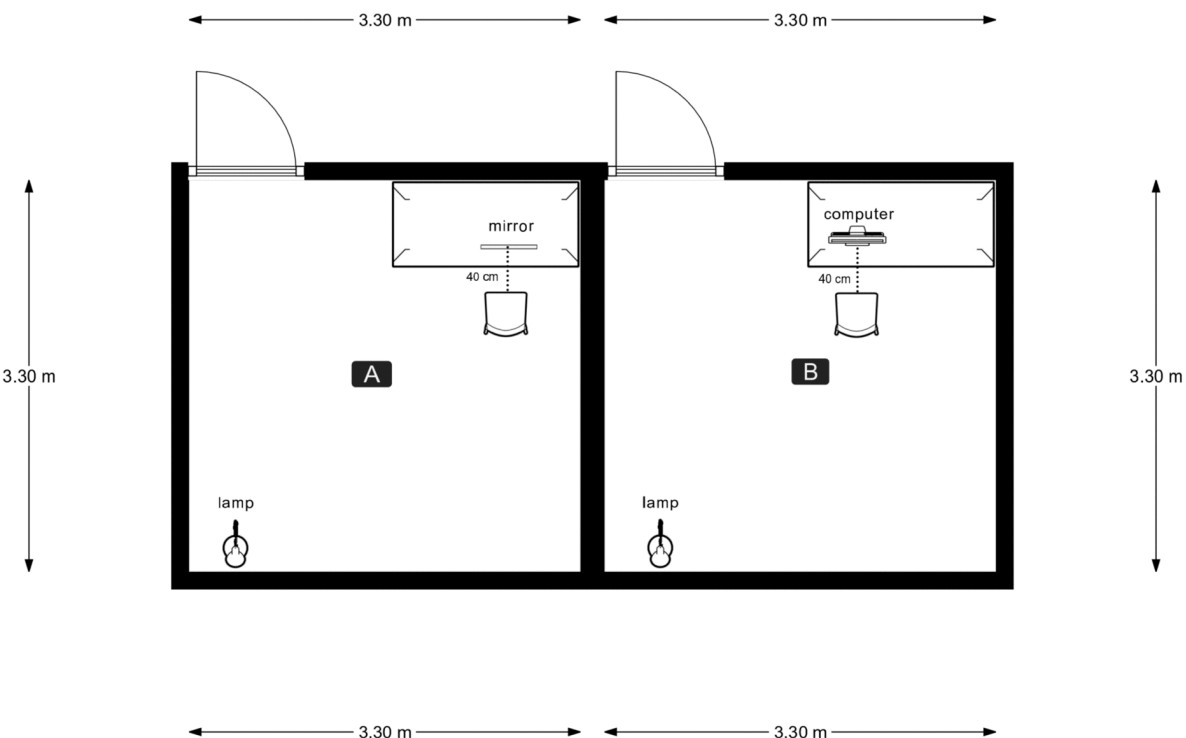

**Fig 1. Setup of the mirror-gazing experimental manipulation.**

eyes, the latter of which was the original instruction described by Caputo [26]. We aimed for minimal instructions to avoid deliberate effort that might hinder the onset of a dissociative experience [46]. In the control condition, a 27.5 x 49 cm computer monitor was placed 40 cm from the participants (See Fig 1, Side B), and they watched a video clip of nature scenery (see [38] for a similar procedure).

All conditions lasted 7 minutes. The mirror paradigm of Caputo [26,27] lasts 10 minutes, but when we conducted an initial test among our lab members some of them reported feeling uncomfortable being in front of the mirror for this time period. Since Miller and colleagues [25] were able to induce dissociative experiences after only three minutes of mirror-gazing, we decided to shorten the procedure. Following the induction, participants re-answered the state questionnaires (T2 measurement). Then, they performed the 10-minutes simple cognitive task and answered the state battery one last time (T3 measurement). Lastly, participants reported demographics and trait dissociation and were debriefed by the experimenter.

### Data analysis

Missing data were estimated for all questionnaire items in each assessment. Out of a total of 139 items, 64 items did not have any missing data, 76 had 1% of missing data, and one item had 4% of missing data. Since missingness under 5% is considered negligible [47], we did not attempt to replace missing values.

Two-way mixed ANCOVAs were used to analyze the influence of dissociation induction on self-reported measures of dissociation (i.e., manipulation checks) and self-reported sense of agency. The between-factor was condition (MG, MGS, V), the within-factor was time (T1 = baseline, T2 = post-manipulation, T3 = post-cognitive-task), and the level of trait dissociation was included as a covariate, to control for differences in trait tendencies to dissociate and to explore interactive effects.

## Study 1 results

The descriptive statistics of Study 1 variables are presented in S1 Table. Notably, the differences in trait dissociation between the groups were not statistically significant.

### Manipulation check

A significant three-way interaction between time, condition, and trait dissociation emerged when predicting AII ($\eta^2_p$ = .08, 90% CI [0.02, 0.14], $F_{(4,182)}$ = 4.19, $p$ = .002), and a marginal interactive effect was detected when predicting DPDR ($\eta^2_p$ = .05, 90% CI [0.00, 0.10], $F_{(4, 182)}$ = 2.29, $p$ = .062). We performed post-hoc multiple comparison tests with the Bonferroni-Holm correction (see Figs 2, 3). Regarding AII, state levels were increased at T2 across almost all conditions and all levels of trait dissociation. Specifically, among average dissociators, AII increased in all conditions ($p$ < .001 for all). A similar pattern emerged among low dissociators ($p$ < .001 for MG, $p$ < .001 for MGS, and $p$ = .002 for V). In contrast, among high dissociators, AII increased only in the MGS and V conditions ($p$ < .001 for both). Regarding DPDR, state levels were increased at T2 in different conditions depending on the level of trait dissociation. Specifically, among average dissociators, DPDR increased in all conditions ($p$ = .001 for MG, $p$ < .001 for MGS, and $p$ = .007 for V). However, among low dissociators, DPDR significantly increased only in the MG condition ($p$ = .001), whereas in high dissociators, only in the MGS condition ($p$ < .001) and the V condition ($p$ = .018). Regarding the persistence of the effects, all significant increases observed from T1 to T2 in the levels of state dissociation (DPDR or AII), decreased between T2 and T3. The post-hoc analyses are available in full in S2 Table.

### Self-reported sense of agency

A significant three-way interaction between time, condition, and trait dissociation emerged when predicting explicit sense of agency ($\eta^2_p$ = .07, 90% CI [0.01, 0.12], $F_{(4, 182)}$ = 3.32, $p$ = .012). Post-hoc comparisons with the Bonferroni-Holm adjustment revealed that the sense of agency significantly decreased following the manipulation only in the MGS condition,

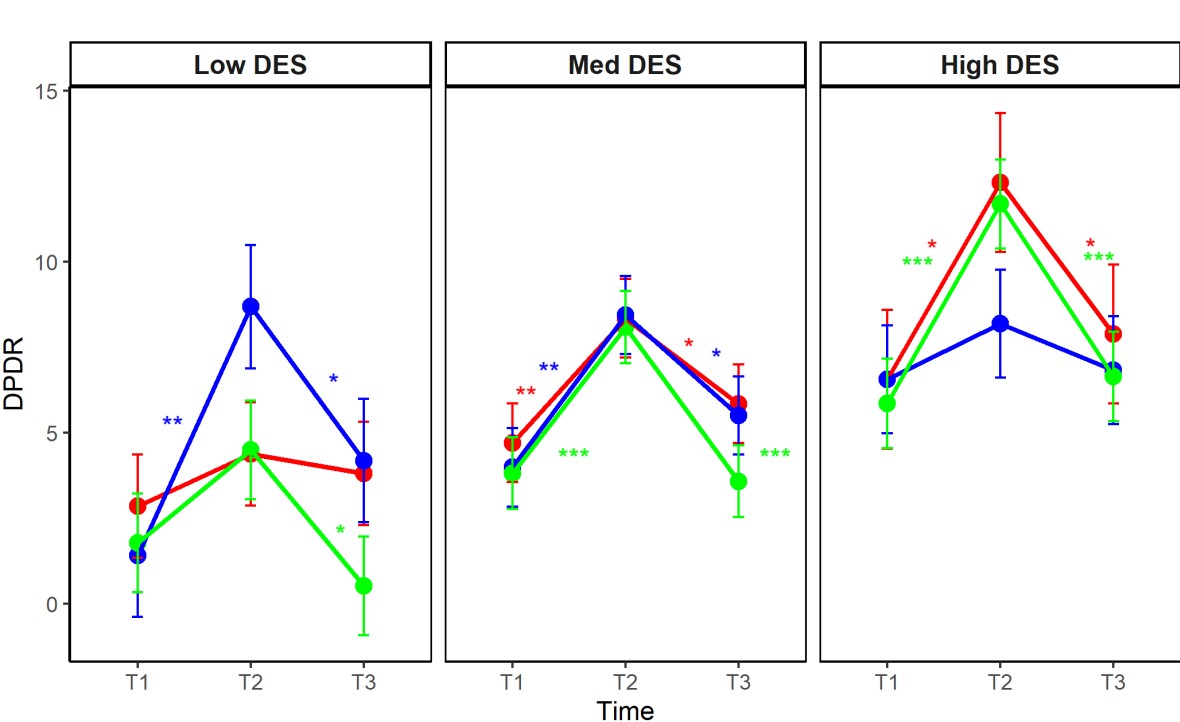

**Fig 2. Post-hoc comparison of changes in state DPDR levels, Study 1.**

among participants with average and high levels of trait dissociation ($p = .002$ for average dissociators, $p < .001$ for high dissociators). These declines did not remain stable but increased significantly between T2 and T3. The post-hoc analysis is presented in Fig 4. The full details of this analysis are available in S2 Table.

Notably, the trait DES contains an item about depersonalization in front of a mirror ("Some people have the experience of looking in a mirror and not recognizing themselves"). Since the trait scales were administered *after* the manipulation, we wanted to rule out the possibility that the manipulation influenced that item, impacting our findings. Therefore, we computed a corrected DES score excluding this item and repeated all analyses involving the DES. The results were unchanged.

## Study 1 discussion

The findings partially supported our hypotheses. The experimental conditions indeed induced dissociation, but so did the control condition. In other words, watching a nature scenery video increased dissociation similarly to gazing at one's reflection. Finding an increase in the control group is a common challenge in studies attempting to induce dissociation [25,28,48–50]. Perhaps narrowing attention to follow experimental demands produces dissociation. The lack of distinction between manipulation and control in this study was mostly evident regarding AII. Since becoming highly immersed while watching a movie is a common description of an AII experience, possibly, watching a video induced AII and was unsuitable as a control condition in this context. Hence, one of the aims of Study 2 was to introduce an improved control condition for proper comparison.

Nonetheless, the study conditions did differ significantly when predicting the sense of agency. Only participants from the MGS condition reported a disrupted sense of agency following the manipulation. Thus, although all conditions

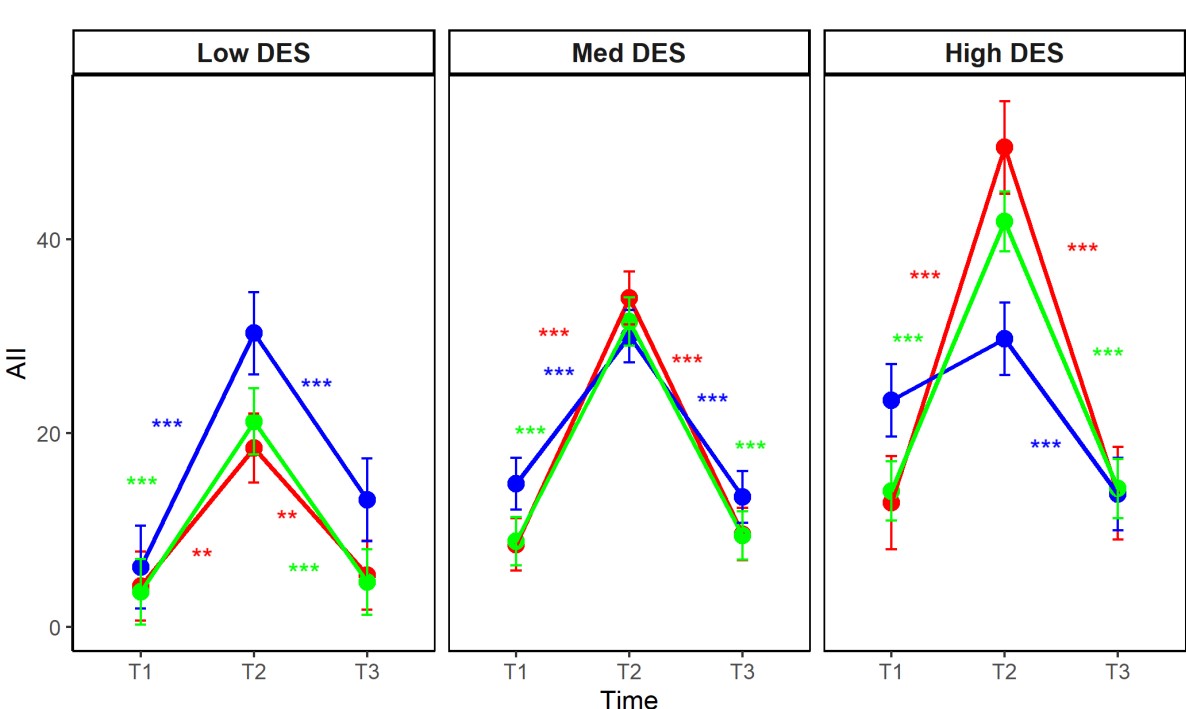

**Fig 3. Post-hoc comparison of changes in state AII levels, Study 1.**

reportedly induced dissociation, this may not have been a uniform process, as only the MGS dissociation was strong enough to reduce agency. The slightly stronger results of the MGS condition are consistent with the claims for a central role of suggestibility in the dynamics of dissociation [30]. Notably, several participants spontaneously shared their surprise at the unusual experiences they had during the debriefing phase with the experimenter, immediately after the assessment. In addition, a recent incidental conversation with a former MGS participant, two years after the fact, led to a vivid description of how she recalls her experience (translated by the first author):

From the start, I felt that I looked strange to myself because of the unusual lighting, and as time passed, I began to feel that my face became distorted and did not look like itself. The instruction was not to move my gaze, so I focused, and as time passed it seemed weirder. The face got more distorted until I could not recognize myself. I was afraid it would get stuck, but it didn't – the distortion effect dissipated quickly once the study was over. But the strange experience didn't. I was left with an uncanny feeling that I can't put my finger on. It lasted for several hours afterwards.

Anecdotal descriptions like this one strengthen the quantitative findings showing the successful effect of the MGS intervention on dissociation induction. Also, as expected, trait dissociation interacted with the effect: sense of agency decreased following the manipulation only among average to high dissociators. In terms of temporality, all effects seem to fade after about 10 minutes, at least according to participants' self-reports. Therefore, we tried to design the intentional binding task of Study 2 to be as short as possible.

A limitation worth mentioning concerning the Study 1 sample is the seemingly higher trait dissociation scores of the experimental groups compared to controls. Although this difference did not reach statistical significance, it may have contributed somewhat to the findings showing an effect in those groups. However, it is important to remember that the trait questionnaire was administered *after* the manipulation, to avoid any bias to the state measurements. Thus, it could have

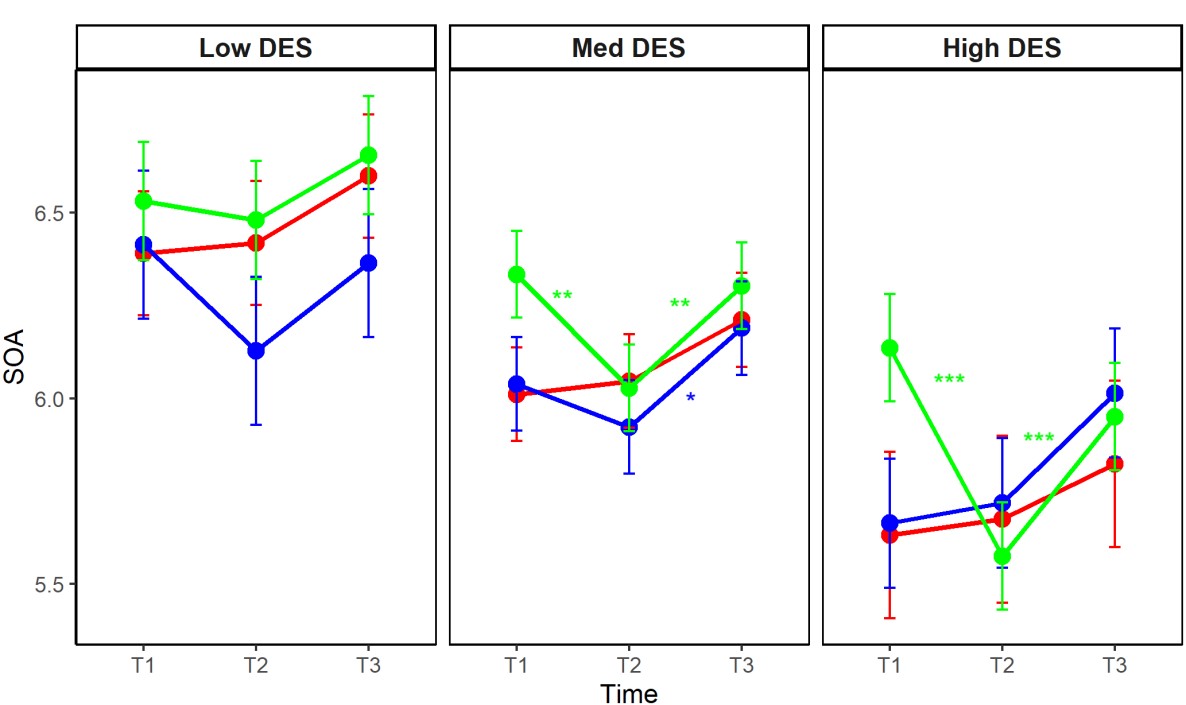

**Fig 4. Post-hoc comparison of changes in state self-report agency levels, Study 1.**

possibly been affected by the manipulation (e.g., those who are more easily suggestible would report that they tend to be dissociative after undergoing either of the experimental manipulations, MG or MGS).

To summarize, Study 1 findings showed that laboratory-induced dissociation may bring about disruptions in self-reported sense of agency. In the next part of this exploration, we relied on our preliminary findings and aimed to design a better-differentiated control condition and assess the sense of agency implicitly. We intended to improve the experimental design by adding a new control condition which would be less likely to elicit dissociative experiences. Since we found that passive staring at a video evoked a feeling of absorption in the visual stimulus, we wanted the participants to have active cognitive engagement. Thus, in the new control condition participants read a short article from an online popular science magazine describing the history of Christian art [51]. As this new control condition was not described in previous literature, we decided to also retain the original, absorption-inducing video-watching control, as another group in the study. We thought that as a manipulation that seems to induce mild, daily dissociation, it would be interesting to examine whether this group would exhibit effects that are similar to the new control group or to the dissociation groups. For our second goal, i.e., an implicit assessment of the sense of agency, we used an intentional binding task. Following these changes in the study design, we hypothesized that (1) dissociation would increase in all groups except for the new control condition; and (2) sense of agency would decrease in all groups except for the new control condition.

## Study 2 method

### Participants

Participants enrolled through the institutional experiment system, between December 19th, 2021, and March 23rd, 2023. As before, the study was approved by the institutional IRB according to the declaration of Helsinki and the invitation to

participate stated that mental health is required, and having eyeglasses was an exclusion criterion. Two-hundred and eight undergraduates participated, 164 for course credit and 44 for monetary compensation of ~10$ (136 females, mean age 23.58 (SD = 1.70)). Nine participants had too many missed trials on the intentional binding task (see the data analysis section below), leaving a final sample of $N = 199$ for statistical analyses. As in Study 1, all participants were native or fluent Hebrew speakers and mostly came from middle-class or upper-middle-class backgrounds.

## Measures

**Questionnaires.** The trait and state questionnaires were identical to those used in Study 1. Internal reliability was.95 for the DES and.85 (T1),.89 (T2),.91 (T3) for the CADSS,.84 (T1),.87 (T2),.90 (T3) for AII, and finally,.85 (T1),.90 (T2),.91 (T3) for the SoAS.

**The intentional binding task.** Intentional binding (IB) is a widely known implicit measure of the sense of agency based on time perception, developed by Haggard et al. [20]. It is based on the finding that in voluntary events, we tend to bind our actions with their outcomes, i.e., to perceive them as closer in time than they actually are. Hence, the higher the sense of agency, the shorter the estimated interval between an action and its effect. In the IB task, participants are instructed to perform an action (key press) that elicits an outcome (a tone) and are asked to estimate either the timing of their action (action-focused trials) or the timing of the outcome (outcome-focused trials). A detailed description of the task we developed for the current study is available in S2 Text.

The task yields implicit and explicit indices of the sense of agency. The implicit ones are measures of binding actions and outcomes, i.e., the judgment errors of estimating the keypress and its consecutive tone as closer than they actually are. We also took into consideration each participant's baseline level of precision in assessing the timing of actions and of auditory stimuli. Thus, each participant had four implicit indices: judgment error regarding the timing of action (keypress), judgment error regarding the timing of outcome (tone), and "Action binding" and "Outcome binding", which are the same judgment errors, minus each participant's baseline level of precision. In all implicit indices, greater deviations from the real timing indicated a higher sense of agency. The explicit indices were participants' answers when they were asked to what extent their keypress influenced the tone, with a higher score indicating a higher sense of agency. In most trials, the keypress indeed resulted in a tone, and this index reflected a benign sense of agency. However, in 25% of the trials, a tone was heard regardless of participants' actions; thus, in these trials, the explicit index reflected an illusory sense of agency.

## Procedure and experimental design

Before arrival, each participant was randomly allocated to one of four study conditions: (1) mirror-gazing (MG) ($N = 48$), (2) mirror-gazing with suggestion (MGS) ($N = 48$), (3) watching a nature scenery video (old control condition from Study 1, termed Video – V) ($N = 53$), (4) reading an article (the new control condition, designed for Study 2, termed Control – C) ($N = 50$). Conditions 1–3 were similar to Study 1. In condition 4 (C), an article about the history of art in Christianity [51] was displayed on the computer monitor, and participants were instructed to read it until stopped by the experimenter.

The consent process and the experimental procedure were identical to Study 1, except participants performed the IB task instead of the Posner task.

## Data analysis

Missing data were estimated for all questionnaire items in all assessments. Out of a total of 139 items, 110 items did not have any missing data. In the remaining 29 items, the rates of missingness ranged between 0.5% to 1.5%. Thus, again, we did not replace missing values.

In the IB task, the first three trials of each condition were defined as practice and were omitted from the analyses. Trials in which participants did not respond, or in which the difference between the real response time and the estimated time was greater than two seconds (which implies that participants did not pay sufficient attention) were also omitted from

analyses. One participant who did not perform the task at all and eight participants with more than 10% of missing trials were excluded from analyses, so the final sample included $N = 199$ participants.

As before, two-way mixed ANCOVA was used to analyze the influence of group allocation on self-reported measures of dissociation (i.e., manipulation checks) and self-reported sense of agency. The between-factor was condition, this time with 4 levels (MG, MGS, V, and C), the within-factor was time (T1 = baseline, T2 = post-manipulation, T3 = post-IB-task), and the level of trait dissociation was integrated as a covariate.

Next, one-way ANOVA was used to examine the effect of dissociation induction on implicit and explicit measures of sense of agency from the IB task. The between-factor was condition. The IB task was administered once, rendering no within-subject time factor. Initially, this analysis was conducted as an ANCOVA with trait dissociation as a covariate, but trait dissociation had no influence on the effect of condition over IB measures, and therefore we present results from a simpler, ANOVA model. Study 2 drew on the findings of Study 1 and aimed to answer more specific questions. Hence, in addition to ANOVA, we defined planned contrasts as follows:

C1: Comparison of C to the mean of all other conditions.

C2: Comparison of V to the mean of the experimental conditions (MG and MGS).

C3: Comparison between experimental conditions (MG versus MGS).

## Study 2 results

The descriptive statistics are detailed in S4 Table. As in Study 1, the differences in trait dissociation between the groups were not statistically significant.

### Manipulation check

A significant three-way interaction between time, condition, and trait dissociation emerged when predicting both dissociation variables (DPDR: $\eta_p^2 = .08$, 90% CI [0.03, 0.11], $F_{(6, 382)} = 5.36$, $p < .001$); AII: $\eta_p^2 = .08$, 90% CI [0.03, 0.11], $F_{(6, 382)} = 5.19$, $p < .001$), implying that different conditions induced different levels of dissociation, and that the strength of the induction was affected by the trait tendency to dissociate. The ANCOVA was followed by post-hoc multiple comparison tests, with the Bonferroni-Holm adjustment (Figs 5, 6). For DPDR, state levels increased at T2 in the experimental conditions, varying according to the level of trait dissociation. Among average and high dissociators, DPDR increased in conditions V, MG and MGS, but not in C ($p < .001$ for all significant changes), supporting the adequacy of C as a control condition for DPDR induction among average and high dissociators. Among low dissociators, state levels did not change significantly in any condition. For AII, state levels increased at T2 in nearly all conditions and levels of trait dissociation. Among average dissociators, AII increased in all conditions ($p < .001$ for V, MG, and MGS; $p = .005$ for C). A similar pattern was observed among high dissociators ($p < .001$ for V, MG, and MGS; $p = .003$ for C). Among low dissociators, AII increased in conditions V, MG and MGS, but not in C ($p < .001$ for MG, $p = .001$ for MGS, and $p = .004$ for V), this time supporting the adequacy of C as a control condition for AII induction among low dissociators. Regarding the persistence of the effects, most of the significant increases in state DPDR remained stable at T3, whereas most of the significant increases in state AII had diminished by T3. The full analyses are detailed in S5 Table.

### Self-reported sense of agency

A significant three-way interaction between time, condition, and trait dissociation emerged when predicting self-reported state sense of agency ($\eta_p^2 = .06$, 90% CI [0.02, 0.09], $F_{(6, 382)} = 4.02$, $p < .001$). Post-hoc comparisons adjusted with the Bonferroni-Holm procedure (Fig 7), revealed that the sense of agency significantly decreased following the manipulation (i.e., between T1 and T2) only among participants with high levels of trait dissociation, in conditions MG ($p = .017$) and C

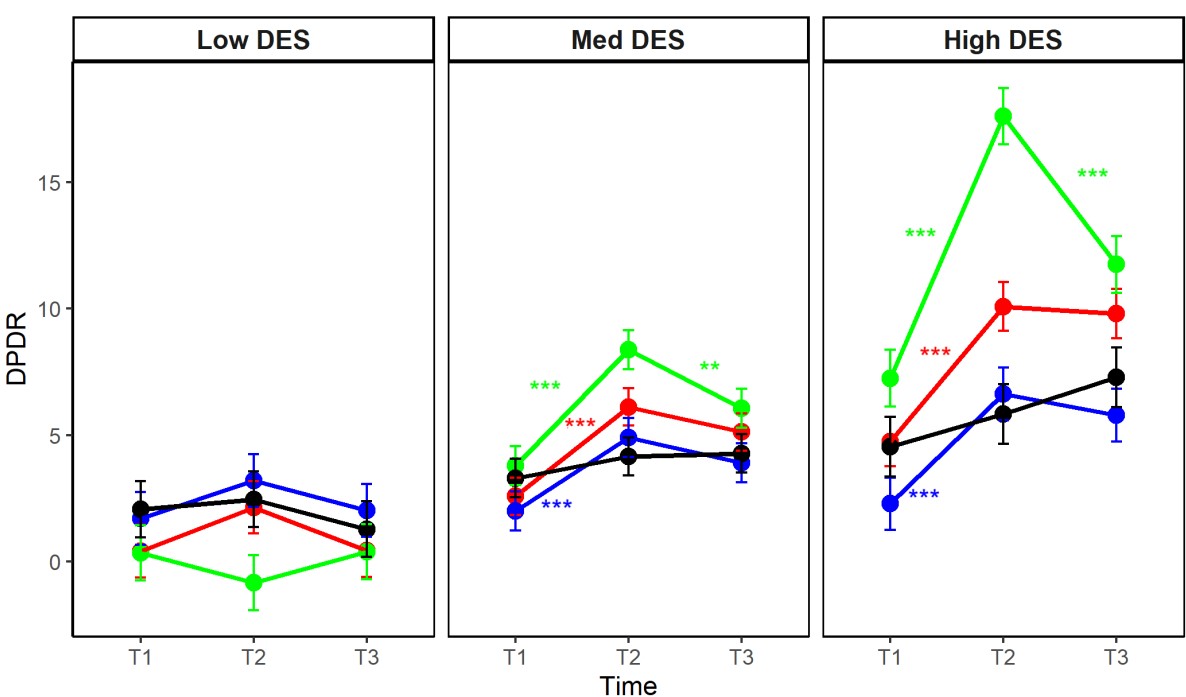

**Fig 5. Post-hoc comparison of changes in state DPDR levels, Study 2.**

($p = .040$), and that decrease remained stable between T2 and T3 only in condition MG. The full analysis details are available in S5 Table.

As in Study 1, we repeated all analyses with a corrected DES score, excluding the mirror item. Again, results were unchanged.

### Intentional binding results

**Implicit measures: Action judgment error and action binding.** When predicting Action Judgment Error, there was a marginally significant effect of experimental condition ($\eta^2_p = .034$, 90% CI [0.00, 0.07], $F_{(3, 195)} = 2.27$, $p = .082$). Controlling for the effect of trait level of dissociation had no impact on this effect. Planned contrasts (see Fig 8) revealed that participants in condition C demonstrated a significantly higher sense of agency (M = 133.43 ms) compared to the mean of all other conditions (101.22, 68.81, and 82.76 ms, in V, MG, and MGS, respectively; $\eta^2_p = .027$, 90% CI [0.00, 0.07], $F_{(1, 195)} = 5.34$, $p = .022$). As in Study 1, V did not differ significantly from the mean of the experimental conditions ($\eta^2_p = .007$, 90% CI [0.00, 0.05], $F_{(1, 195)} = 1.30$, $p = .255$). The two experimental conditions were also not significantly different on this type of agency ($\eta^2_p = .001$, 90% CI [0.00, 0.02], $F_{(1, 195)} = 0.28$, $p = .600$).

When predicting Action Binding, there was no significant effect for condition ($\eta^2_p = .015$, 90% CI [0.00, 0.04], $F_{(3, 194)} = 0.99$, $p = .399$). Again, controlling for the level of trait dissociation did not impact the results. In planned contrasts analysis, the comparison of C to the rest of the conditions was marginally significant in the direction of the hypothesis ($\eta^2_p = .015$, 90% CI [0.00, 0.05], $F_{(1, 195)} = 2.93$, $p = .088$), but the other two planned contrasts were not significant.

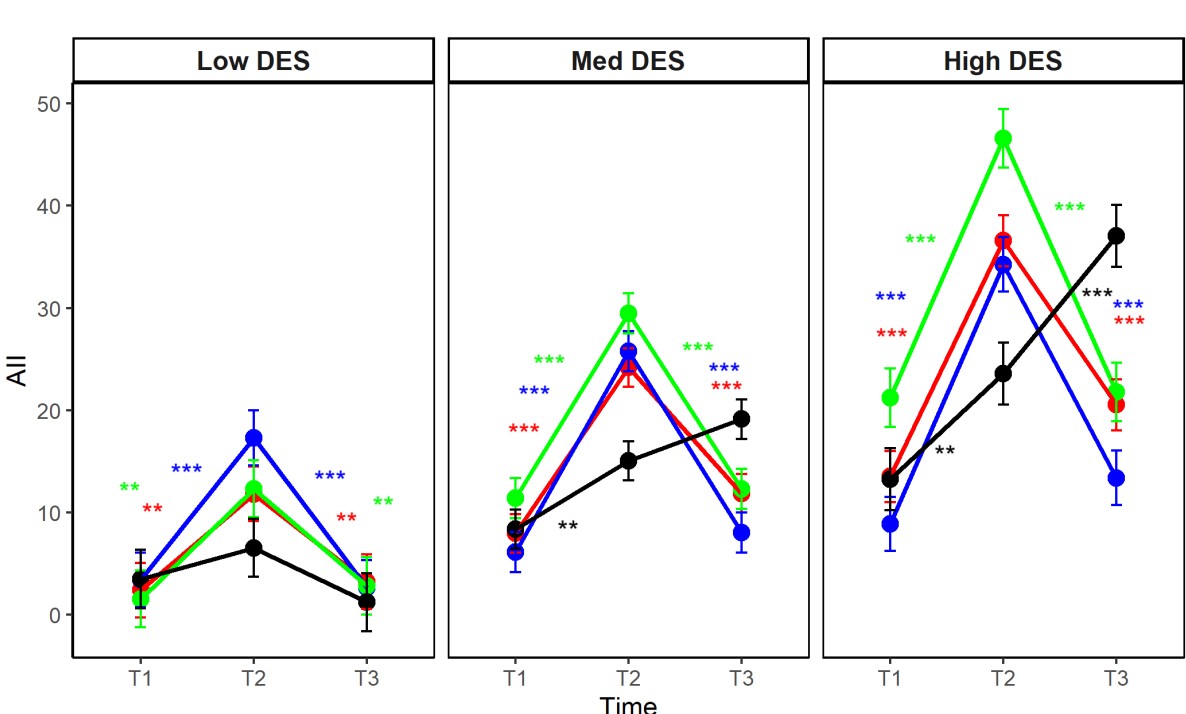

**Fig 6. Post-hoc comparison of changes in state All levels, Study 2.**

**Implicit measures: Outcome judgment error and outcome binding.** No significant effect of condition on Outcome Judgment Error or Outcome Binding was detected, even when controlling for trait level of dissociation. Planned comparisons did not reveal any significant effects either.

**Explicit task-related agency measures.** There was no significant effect of condition on the explicit rating of agency in the action-focused trials, even when controlling for trait level of dissociation. Planned comparisons did not show any significant effects either. The explicit rating of agency in the outcome-focused trials was also not significantly predicted by condition, even when controlling for trait level of dissociation. Planned comparisons revealed a marginally significant difference between V and the mean of the experimental conditions (MG + MGS) ($\eta^2_p$ = .016, 90% CI [0.00, 0.05], $F_{(1, 195)}$ = 3.14, $p$ = .078). The rest of the planned comparisons did not display any significant effects. There was also no effect of condition on the ratings of illusory sense of agency.

## Study 2 discussion

The findings of Study 2 mostly supported our hypotheses. As expected, participants in the new control condition C demonstrated significantly less dissociation after the manipulation compared to participants in all other conditions. Thus, the new experimental setup improved the distinction between the control and the experimental conditions and enabled the detection of the expected effects.

Perhaps most importantly, participants in condition C exhibited a higher implicit sense of action agency than in the other conditions, suggesting that the manipulated dissociative effects decreased the sense of agency. It is worth mentioning that our participants perceived their actions as drifting towards their outcomes, but not the other way around. This finding aligns with a meta-analysis on the IB task, which concluded that action error judgements depend on whether participants

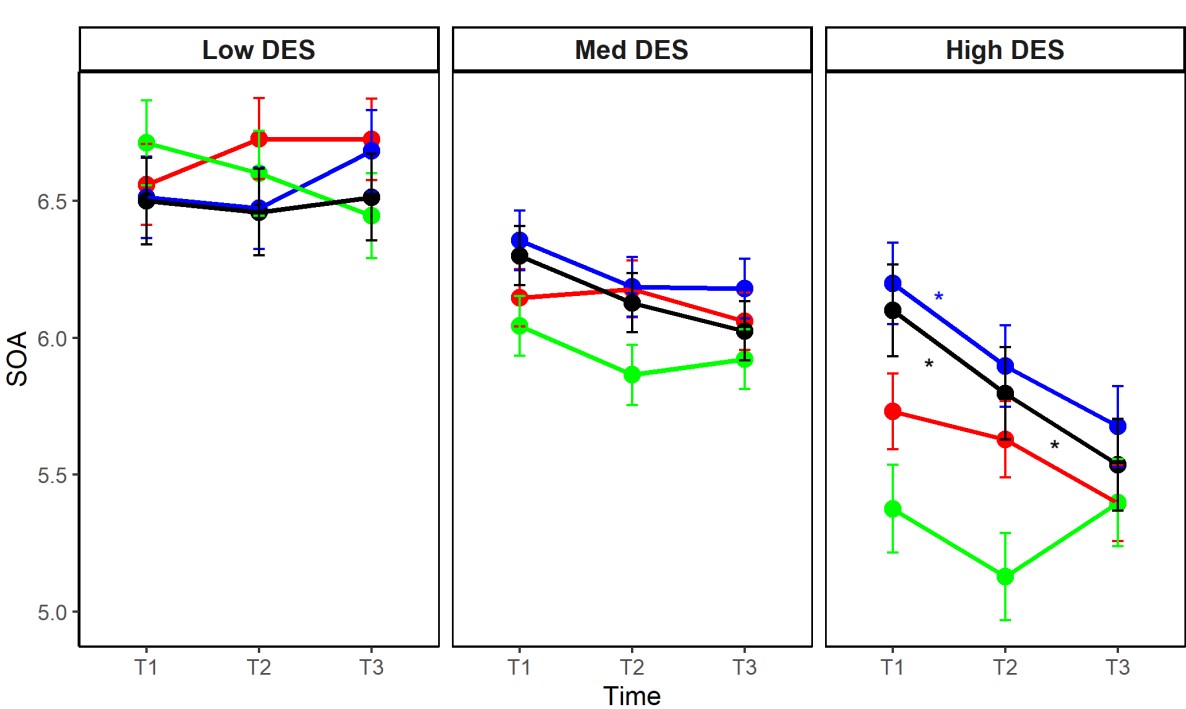

**Fig 7. Post-hoc comparison of changes in state self-report agency levels, Study 2.**

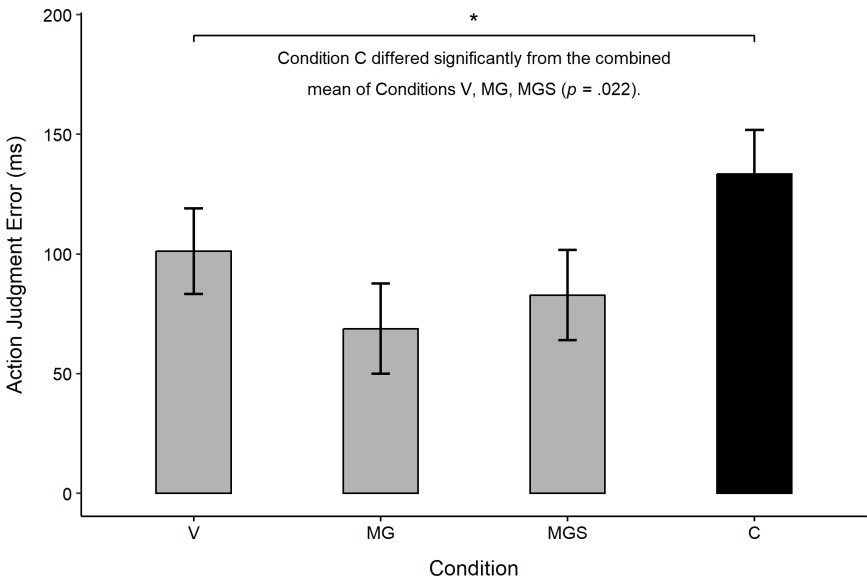

**Fig 8. Planned contrasts of Action Judgment Error across experimental conditions, Study 2.**

can control the onset of outcomes with their actions, whereas outcome error judgements occur when participants are asked to compare predicted and observed outcomes [52]. As our participants initiated the outcome sounds, and the task did not require outcome prediction, it is not surprising that only action effects were observed.

Notably, when self-reported sense of agency was examined, both participants of the new control condition and of the experimental MG condition reported a decreased sense of agency following the manipulation. This was despite encouraging manipulation check results (i.e., dissociation was increased differentially in accordance with the hypothesis). This means that the increase in dissociation, occurring only in the experimental groups, cannot fully explain the decrease in agency, which occurred also in the control group. Perhaps, the measure we used for self-reported sense of agency was not suitable for the current research settings. This index was originally developed as a measure of trait sense of agency [18]. Our team had previously used a modified version of it to examine state levels of the sense of agency [7,8], but it had not been included in experimental designs until now and thus had not been used to examine changes in the sense of agency following manipulations.

Individual differences in the tendency to dissociate contributed to the induction of DPDR experiences and to the disruption of self-reported sense of agency, with average or high dissociators reporting stronger effects, similar to Study 1. However, trait individual differences did not affect the implicit effects of the sense of agency. People who report that they are often prone to dissociation may be more sensitive to unusual consciousness experiences, or more biased to respond positively to direct questions about such experiences, and thus were more likely to answer affirmatively through self-report. By measuring the sense of agency implicitly, we were able to bypass self-report styles and achieve a different, perhaps "purer", expression of altered consciousness. This may explain why all participants in conditions MGS, MG, and V had decreased implicit action agency, even though only average and high dissociators self-reported an increase in DPDR in the manipulation checks. Low dissociators may have also experienced changes in perception but were unaware of them and could not report them. This underscores the importance of using implicit measures alongside self-report measures.

The decision to change the control condition in Study 2 compared to Study 1 was also supported, as the dissociation-inducing effects of the V condition emerged in this study as well. Also noteworthy is that in the present study, the difference in trait dissociation between the groups was again non-significant, and this time, it was also much more minimal, even though it was still measured after the manipulation. Lastly, the slightly better performance of the MGS condition observed in Study 1 was not replicated. This condition did not provide an advantage compared to other experimental conditions, and thus, we cannot definitively say that using suggestions is necessary for efficiently inducing dissociation. This finding aligns with recent research on the role of suggestion in the mechanism underlying the mirror-gazing paradigm [32].

## General discussion

These findings provide experimental evidence that dissociative states lead to a decrease in the sense of agency. A previous investigation found that a similar paradigm induced an impaired sense of agency during the mirror-gazing manipulation itself [33]. However, this prior study fused dissociation with disruptions of agency, both in the manipulation and in the measurement, preventing the conclusion of causality. The present study supports previous findings and enhances them by enabling the inference of causality, as we manipulated dissociation under controlled conditions and measured dissociation and agency separately. Thus, we have demonstrated that dissociation affects agency. Furthermore, our findings confirm that the causal relationship is evident both when measured explicitly and implicitly.

It seems that a coherent sense of self may be a prerequisite for perceiving the self as actively engaged in its surroundings. This basic process may represent a broader dynamic where dissociative experiences may hinder people's ability to plan, initiate, act, and create in their surroundings. This echoes recent research showing that complex trauma (which could be construed as trauma that deeply affected the ongoing sense of self), more than simple trauma, was associated with reduced daily sense of agency [8]. Difficulty in attributing agency to one's actions may presumably partially explain why dissociation predicts poorer response to psychotherapy [53–55]. A causal effect of the sense of self on agency may

also be relevant to understanding psychosis. For example, Schizophrenia is considered by several researchers to represent a disorder of the sense of self [56–58], and it is indeed characterized by a breakdown in the sense of agency [59–61].

The impairment in the sense of agency following dissociative experiences may help explain the dynamics of borderline personality disorder (BPD) and bulimia nervosa. Dissociative experiences are relatively common among individuals with BPD [1] and have been found to be associated with the frequency and intensity of binge-eating episodes [62,63]. Notably, both bulimia and BPD have recently been linked to a disrupted sense of agency [24,64–66]. Thus, the causal relationship identified in our investigation suggests that dissociative experiences may contribute to agency disruptions in these disorders, potentially explaining some of their behavioral manifestations. The causal relationship identified in the current investigation may carry additional clinical implications. There is evidence that the state level of agency can affect memory, as having a sense of agency over an action was found associated with better memory for items involved in that action [67]. Thus, perhaps DPDR reduces agency, which in turn impairs memory encoding. Such a causal chain may underlie the connection between experiences of dissociation, decreased agency, and vague memories, a connection which is relevant to the debate regarding dissociation and memory problems [68]. The sense of agency is also recognized as a developmental resilience factor, which enables the establishment of social and emotional well-being both in childhood [69] and young adulthood [70]. Frequent experiences of dissociation, inducing impaired agency, may hinder achieving essential developmental milestones.

The results provide additional support for the literature on mirror-gazing as a viable method for inducing dissociation [26,27,38,71,72]. The procedure of mirror-gazing resembles compulsive-like staring, a process that was found to increase dissociative experiences [73]. Obsessive-Compulsive Symptoms (OCS) are highly relevant to both of our constructs of interest, as OCS were found related to an impaired sense of agency [19,74,75], as well as to dissociative tendencies [76]. The current study did not evaluate OCS, but it may contribute to understanding the relationship between OCS, dissociation, and alterations in the sense of agency, as it corresponds to relevant theoretical models. In a recently proposed model [76], the link between OCS, agency, and dissociation was explained through an underlying mechanism of inward-focused attention. Relatedly, another theoretical framework suggested that dissociative symptoms and impaired sense of agency rely on an excessive allocation of attention to internal stimuli [11]. These models are supported by the current research, which demonstrates that attention directed constantly at one's own reflection in a compulsive-like manner can disintegrate the experience of being (i.e., increase dissociation), and erode the relationship between action and its outcome (i.e., impair the sense of agency).

Another finding of the current research was that trait levels of dissociation affected the efficacy of the manipulation on self-report outcomes. Specifically, trait dissociation seemed to enhance the ability of the mirror-gazing task to increase self-reported dissociation and decrease self-reported levels of agency. This supports a recent hypothesis set whereby following a self-focusing task, high dissociators will experience a decreased sense of agency [11]. Interestingly, however, trait dissociation did not influence the effect of the manipulation on the implicit measurement of the sense of agency. This discrepancy between direct and implicit evaluation of the sense of agency corresponds with previous findings [19,74,75,77] and reinforces the decision to include both types of measures in the study. It seems that when people are asked directly about their sense of agency, they are biased by the way they perceive themselves or by socially acceptable answers, while an implicit assessment of the sense of agency is free of such bias. Implicit measurement in this study suggested that, on average, individuals were affected by the manipulation even if they did not report an elevation in state dissociation and reported that they do not tend to have such experiences as a trait.

## Limitations and constraints on generality

Several limitations of the current investigation should be mentioned. First, as mentioned before, trait dissociation was measured at the conclusion of the experiments. Even though the DES is a well-validated questionnaire, which was designed to assess a stable tendency, it is possible that participants' answers to the DES were influenced by the experimental condition they were assigned to, or by their pattern of answers to the state questionnaires throughout the entire

experiment. Although we made sure that any state effect on the trait mirror item did not affect the results, it would probably have been better to randomize the timing of the administration of the DES, or to ask participants to answer it on a different day. Second, considering the decrease in the self-reported measures between T2 and T3, it is possible that the induced consciousness experiences were largely transient and not sufficiently present when participants performed the IB task. Although some of the effects found in this task supported our hypotheses, it is possible that the transient nature of the effects of the manipulation rendered the study underpowered, preventing us from detecting additional effects or stronger ones. Possibly, an identical replication of the Caputo mirror-gazing paradigm [26], with 10-minute gazing, a symmetrical laboratory setup, lower face illumination and direct instruction to focus on the eyes, would have yielded stronger, more stable effects. Moreover, supervising the correct execution of the task by placing a micro-camera on the surface on which participants were instructed to gaze, might have strengthened the task effect. This is relevant given the participant's remark cited in the discussion of Study 1, which may allude to an inclination to look away from the mirror. Also, we may have limited our ability to fully characterize what our participants experienced during mirror gazing, as we did not use a dedicated measure for it. Using the Strange Face Questionnaire [32] would perhaps provide additional insight into the dissociation and agency data we collected. Finally, our sample was a student sample, which was demographically relatively homogeneous and highly functioning. Thus, our findings might not be generalizable to clinical populations. To increase external validity, future studies should include more diverse samples, especially in terms of age and clinical status.

## Conclusions

Despite these shortcomings, this study has notable strengths. Its innovative experimental design, which is scarce in the dissociation research field, provides novel causal evidence for the dynamics of dissociation and disruptions in the sense of agency. In addition, the combination of direct and indirect measures evaluating the sense of agency contributes to the robustness of the findings and elucidates the discrepancy between participants' subjective experience of themselves, and their actual performance in the world.

## Supporting information

**S1 Text. The mirror-gazing paradigm.**
(DOCX)

**S2 Text. The intentional binding task.**
(DOCX)

**S1 Table. Descriptive data of Study 1 variables.**
(DOCX)

**S2 Table. Post-hoc contrasts predicting self-reported state dissociation and sense of agency, controlling for trait dissociation (Study 1).**
(DOCX)

**S3 Table. The intentional binding task: instructions and events per condition.**
(DOCX)

**S4 Table. Descriptive data of Study 2 variables.**
(DOCX)

**S5 Table. Post-hoc contrasts predicting self-reported state dissociation and sense of agency, controlling for trait dissociation (Study 2).**
(DOCX)

**S1 Fig. The intentional binding clock.**
(TIF)

## Author contributions

**Conceptualization:** Noa Bregman-Hai, Nirit Soffer-Dudek.

**Data curation:** Noa Bregman-Hai.

**Formal analysis:** Noa Bregman-Hai.

**Investigation:** Noa Bregman-Hai.

**Methodology:** Noa Bregman-Hai.

**Project administration:** Noa Bregman-Hai.

**Software:** Noa Bregman-Hai.

**Supervision:** Nirit Soffer-Dudek.

**Validation:** Noa Bregman-Hai.

**Writing – original draft:** Noa Bregman-Hai.

**Writing – review & editing:** Noa Bregman-Hai, Nirit Soffer-Dudek.

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
