## [Decision Letter · Decision Letter 0]

10 Sep 2025

Dear Dr. Bregman-Hai,

Thank you for submitting your manuscript to PLOS ONE. After careful consideration, we feel that it has merit but does not fully meet PLOS ONE’s publication criteria as it currently stands. Therefore, we invite you to submit a revised version of the manuscript that addresses the points raised during the review process.

We look forward to receiving your revised manuscript.

Kind regards,

Clare Eddy

Academic Editor

PLOS ONE

Journal Requirements:

Reviewers' comments:

Reviewer's Responses to Questions

**Comments to the Author**

1. Is the manuscript technically sound, and do the data support the conclusions?

Reviewer #1: Partly

Reviewer #2: Yes

2. Has the statistical analysis been performed appropriately and rigorously?

Reviewer #1: Yes

Reviewer #2: Yes

3. Have the authors made all data underlying the findings in their manuscript fully available?

Reviewer #1: Yes

Reviewer #2: Yes

4. Is the manuscript presented in an intelligible fashion and written in standard English?

Reviewer #1: Yes

Reviewer #2: Yes

Reviewer #1: The manuscript investigates the relevant question of the causal relationship between "dissociative detachments" and sense of agency, which has considerable clinical implications. It should be noted that, in their manuscript, the effect on agency is measured "after" the dissociative task, and not "during" dissociation. Conversely, this last question remains an open endeavor that still has to be investigated. The experimental results of the manuscript may prove interesting.

However, the authors should address important issues before their manuscript can be published. Issues concern both their technical/methodological errors and theoretical considerations.

Major issues:

Technical neglects include the lack of information about the illumination level of the face in front of the mirror. The authors do not report any data about the type of illuminator (LED, halogen, or tungsten, which have entirely different spectral distributions), the power of the lamp, and the face illuminance level. Conversely, facial illuminance is crucial for obtaining dissociative states and "dissociative detachments" during mirror-gazing. One can easily and inexpensively make these simple measurements through a low-cost photometer. The authors should report these data in their revision.

A second technical error concerns the setup of the laboratory (Figure 1). The placement of both the participant and the illuminator is totally asymmetrical. This situation is unsuitable for checking perceptual "dissociative detachments" since the participant cannot distinguish between biased perceptions of reality (produced by asymmetric facial illuminance) and the perception of dissociative faces that they can see in (or beyond) the mirror.

The authors can overcome this issue in their future experiments by (1) placing the mirror in the exact geometrical center of the laboratory, (2) assuring that the wall reflected in the mirror (i.e., "behind" the participant) should be completely empty (which is not the case in the setup of Figure 1), (3) the illuminator lamp should be placed on the floor behind the participant and aligned on the symmetry axis of the laboratory, and (4) the facial illuminance should be about 1 lux or less when using a tungsten or halogen lamp.

For instance, avoiding any suggestion or artifacts in facial illumination due to physical asymmetry in lighting is especially crucial when investigating brain self-generated dissociative detachments of chimeric faces through split-mirror gazing (Caputo, 2021, cited). However, suppose the authors aim to use a "more effective than the standard mirror-gazing paradigm" (line 593): in that case, the authors should first correct their errors, as indicated above, because the correct standard setup is much more effective than they ascertain in the two experiments of their manuscript.

A third technical error concerns task instructions regarding the mirror-gazing paradigm. The participant should be asked to stare at their own eyes reflected in the mirror and not just vaguely gaze at their face.

A fourth technical error is that the experimenter had no means to control for the correct execution of the mirror-gazing task. Indeed, "the participants sat alone in a dimly lit, private inner room" (line 205) with the "experimenter waiting in the larger room outside" (line 206). Therefore, the experimenter cannot control whether the participants directed their gaze correctly toward the mirror, at its frame, or away from the mirror. This impediment is a major drawback of the manuscript and may explain the tiny or absent difference between mirror-gazing and watching a naturalistic video in dissociation or absorption scores. The authors can overcome this crucial issue in future experiments by adopting the technical solution described in an article (Caputo, 2023, Journal Trauma Dissociation, not cited) by placing a small camera in the center of the mirror (with the face visible through approx 3 millimeter hole in the mirror) or just above the mirror.

These errors, which have been listed above, may have caused the low/absent dissociative states measured at T2 after the mirror-gazing task.

The authors should declare these errors in a sub-chapter within the General Discussion of their main manuscript. This sub-chapter should be entitled "Limitations" and placed toward the end of their revised paper. These declarations might exclude (or reduce) the likelihood of persevering into similar flawed executions of the mirror-gazing procedure and setup.

The face conveys one's identity. Hence, mirror-gazing certainly involves both face perception and identity recognition. Therefore, limiting the effect of prolonged mirror gazing only to "dissociative detachments" is incorrect. A famous paper on dissociative states (Holmes et al, 2005, Clinical Psychology Review, not cited) distinguished "dissociative detachments" and "compartmentalizations" of other identities. The authors should update and correct their incomplete theoretical approach to mirror-gazing.

The measure (CADSS) used by the authors for DPDR is inadequate when evaluating mirror-gazing dissociation since it contains no items concerning mirrors. At the same time, DES has only one item dedicated to mirror gazing, and this item could explain the effect of "trait dissociation" that the authors found with various statistical interactions, given that DES (i.e., measurement of trait dissociation) was administered "after" the dissociative task (line 307). In the MGS group, the "suggestion" of paying attention to the face probably did not introduce a bias, as the authors suppose. Instead, it may enhance awareness of face changes in participants who, as a consequence, scored higher on DES.

Supplementary materials:

The authors only mention early explanations of mirror-gazing "dissociative detachments" and omit more recent theoretical accounts. It should be noted that these "dissociative detachments" are specifically linked to face stimuli, thus leading to identity discontinuities since the face affords one's identity. Therefore, from the very beginning, these "dissociative detachments" of one's face can exclude the simple explanation based on the Troxler effect. All three dissociative states (i.e., derealization, depersonalization, and dissociative identity) may be involved in "dissociative detachments" [and compartmentalization of new strange identities]. Indeed, Caputo (2023, Journal Trauma Dissociation, not cited) adopted this theoretical perspective and found that the CADSS scale is not an optimal scale for measuring mirror-gazing "dissociative detachments" [and identity compartmentalization] (indeed, this CADSS inadequacy is confirmed by the description of unsettling experiences of mirror gazing by one participant: lines 290-296).

Finally, the latest structural equation model that investigates "dissociative detachments" [and identity compartmentalization] (Lange et al, 2021, Psychology of Consciousness, not cited) found empirical evidence that the three layers of dissociative states (i.e., 1. derealization, 2. depersonalization, and 3. dissociative identity) are linked together in that exact sequence by causal relationships during eye-to-eye gazing, i.e., the inter-personal version of mirror-gazing of oneself.

Minor issues:

Responses to CADSS items are not on Likert scales but on 5-level scales.

Reviewer #2: The present study examined the causal relationship between dissociative symptoms and sense of agency under controlled laboratory conditions. With two experimental studies, authors described a causal link between those two aspects employing both explicit and implicit measures.

INTRODUCTION

In the introduction section, the authors briefly describe explicit and implicit measures of sense of agency (lines 79-86). I suggest expanding the description of the experimental paradigm previously employed to implicitly assess the sense of agency. Specifically, the authors refer only to the intentional binding paradigm, but they could also describe alternative paradigms such as the sensory attenuation paradigm, which has been successfully employed to investigate alterations of sense of agency also in Borderline Personality Disorder, in which patients often manifest dissociation symptoms (see for examples: https://doi.org/10.3389/fpsyt.2020.00449,
https://doi.org/10.1016/j.neuroimage.2021.117727,
https://doi.org/10.1523/JNEUROSCI.1694-16.2016,
https://doi.org/10.17605/OSF.IO/SZTEK,
https://doi.org/10.1111/ejn.13931,
https://doi.org/10.1038/s41598-023-41345-5).

STUDY 1 RESULTS section

I suggest moving Table 1 to the supplementary materials and replace it with a meaningful graphical representation of the significant interactions emerging from the mixed ANCOVAs on Manipulation check and Self-reported sense of agency. Furthermore, even if full details of post-hoc analyses are provided in the supplementary materials, when describing the results of Bonferroni-Holm post-hoc test, authors should also report the p-values, at least those of significant comparisons. It would help the reader to have an immediate grasp of the statistical significance, thereby increasing the strength of the results.

STUDY 2 RESULTS section

As for the study 1, I again suggest moving Table 2 to the supplementary materials and replace it with a meaningful graphical representation of the significant results of manipulation check, self-reported sense of agency and intentional binding. Even in this case, please provide p-values of significant post-hoc comparisons.

GENERAL DISCUSSION

In the general discussion, authors discussed clinical implications of their work, reporting the cases of Schizophrenia and OCS as related to impaired sense of agency. I suggest expanding this part by also mentioning other psychiatric conditions in which dissociative symptoms are experienced, such as Borderline Personality Disorder and Eating Disorders with a bulimic variant, where alterations of sense of agency have also been previously described.

**Do you want your identity to be public for this peer review?** For information about this choice, including consent withdrawal, please see our Privacy Policy

Reviewer #1: No

Reviewer #2: **Yes:** Marcella Romeo

---

## [Author Response · Author response to Decision Letter 1]

9 Nov 2025

Response to Reviewers

I thank both reviewers for the attention they dedicated to this manuscript. Their comments highlighted areas in the text that were not clear enough, and I believe the revised version is noticeably improved. My answers to the specific comments are below.

Reviewer #1:

The manuscript investigates the relevant question of the causal relationship between "dissociative detachments" and sense of agency, which has considerable clinical implications. It should be noted that, in their manuscript, the effect on agency is measured "after" the dissociative task, and not "during" dissociation. Conversely, this last question remains an open endeavor that still has to be investigated. The experimental results of the manuscript may prove interesting.

However, the authors should address important issues before their manuscript can be published. Issues concern both their technical/methodological errors and theoretical considerations.

Major issues:

1. Technical neglects include the lack of information about the illumination level of the face in front of the mirror. The authors do not report any data about the type of illuminator (LED, halogen, or tungsten, which have entirely different spectral distributions), the power of the lamp, and the face illuminance level. Conversely, facial illuminance is crucial for obtaining dissociative states and "dissociative detachments" during mirror-gazing. One can easily and inexpensively make these simple measurements through a low-cost photometer. The authors should report these data in their revision.

Response:

Thank you for the important technical comment. Designing the experimental procedure, we attempted to follow Caputo's rationale [1, 2] and therefore used a 25W incandescent bulb, as described in those articles. This is now mentioned in the Method section of Study 1 (lines 220-221). Additionally, following your comment, we now checked the illumination level of the face of the person sitting in the room under these conditions with a photometer. The result was between 3-5 lux. Although this is higher than the illumination level in Caputo’s studies, it suggests that higher lux levels may also produce dissociative responses, as reflected by the significant pre-post results of our two studies. Notably, studies other than Caputo’s have not reported lux levels; this is mentioned in a recent review [3]). We agree that reporting them better promotes clinical science; hence, we now describe our lux level in the Method section of Study 1 (lines 226-230) and also address it as a possible limitation in the General Discussion section (lines 643-646).

2. A second technical error concerns the setup of the laboratory (Figure 1). The placement of both the participant and the illuminator is totally asymmetrical. This situation is unsuitable for checking perceptual "dissociative detachments" since the participant cannot distinguish between biased perceptions of reality (produced by asymmetric facial illuminance) and the perception of dissociative faces that they can see in (or beyond) the mirror. The authors can overcome this issue in their future experiments by (1) placing the mirror in the exact geometrical center of the laboratory, (2) assuring that the wall reflected in the mirror (i.e., "behind" the participant) should be completely empty (which is not the case in the setup of Figure 1), (3) the illuminator lamp should be placed on the floor behind the participant and aligned on the symmetry axis of the laboratory, and (4) the facial illuminance should be about 1 lux or less when using a tungsten or halogen lamp. For instance, avoiding any suggestion or artifacts in facial illumination due to physical asymmetry in lighting is especially crucial when investigating brain self-generated dissociative detachments of chimeric faces through split-mirror gazing (Caputo, 2021, cited). However, suppose the authors aim to use a "more effective than the standard mirror-gazing paradigm" (line 593): in that case, the authors should first correct their errors, as indicated above, because the correct standard setup is much more effective than they ascertain in the two experiments of their manuscript.

Response:

Following this comment, we understood that we had not described the procedure in sufficient detail. We designed it with the intention of keeping the wall behind the participant empty. The lighting was placed on the floor and not on the wall. We now realize that this is not clear from a top-view illustration, and therefore we note it in the manuscript (lines 221-224). The lamp was not placed directly behind the participant, because in such a position it would have been reflected in the mirror. Therefore, it was placed in the back corner of the room. As we noted in response to the previous comment, we measured the intensity of the light illuminating the participant’s face, which was between 3-5 lux. We now present all these additional details more clearly in the manuscript.

Regarding the effectiveness of the paradigm, the sentence cited above (regarding the split mirror paradigm possibly being more effective) was erased. Additionally, we now address in more detail the differences between our procedure and Caputo's procedure in the limitations paragraph of the General Discussion (lines 643-646).

3. A third technical error concerns task instructions regarding the mirror-gazing paradigm. The participant should be asked to stare at their own eyes reflected in the mirror and not just vaguely gaze at their face.

Response:

We followed the instructions detailed in “Strange-face-in-the-mirror illusion” [2], in which the task was to gaze at the reflected face in the mirror using a 25W incandescent light bulb.

We debated whether to specifically instruct participants to focus on their eyes or to gaze more generally at their face. One article by Caputo from 2010 [1] specifically mentions the eyes, while the second, shorter article from the same year [2] does not mention this instruction, but rather refers more generally to gazing at one’s face as reflected in a mirror. We decided to adopt the less focused instruction, assuming that a focused instruction may hinder the formation of a dissociative experience [4]. In any case, we now mention this difference between our protocol and Caputo’s in Study 1 Method section (lines 234-236) and in the general discussion (line 645).

4. A fourth technical error is that the experimenter had no means to control for the correct execution of the mirror-gazing task. Indeed, "the participants sat alone in a dimly lit, private inner room" (line 205) with the "experimenter waiting in the larger room outside" (line 206). Therefore, the experimenter cannot control whether the participants directed their gaze correctly toward the mirror, at its frame, or away from the mirror. This impediment is a major drawback of the manuscript and may explain the tiny or absent difference between mirror-gazing and watching a naturalistic video in dissociation or absorption scores. The authors can overcome this crucial issue in future experiments by adopting the technical solution described in an article (Caputo, 2023, Journal Trauma Dissociation, not cited) by placing a small camera in the center of the mirror (with the face visible through approx 3 millimeter hole in the mirror) or just above the mirror.

Response:

In many psychological experiments, we rely on participants’ active participation, yet we do not monitor it, because our presence or the presence of a camera might influence the experimental procedure and will surely complicate it. For example, most studies using computerized tasks do not examine with a camera or eye-tracker whether participants followed the instruction to look at a target object. Such a procedure of including a camera in computerized tasks is not standard in the field of Psychology. We believe that the presence of a camera may affect participants’ responses to the paradigm, and perhaps even disrupt the dissociation induction process. Moreover, we do not think that not having an experimenter in the room should be necessarily considered as an error, as experimenter presence also has downsides (i.e., each approach has advantages and disadvantages). Experimenter presence for assuring the gaze direction of the participant is also quite subjective, and the constant gaze of the experimenter in charge of such a task towards the participant could interact with the experiences of the participant. In the present case, if participants did not look at the mirror at all, we would not expect any increase in the level of dissociation in any condition. Since the level of dissociation increased significantly pre- to post-manipulation, it seems that overall, participants cooperated with the task. The lack of a satisfactory difference between the experimental and control conditions most likely stems from the choice of an inappropriate control condition in Study 1.

5. These errors, which have been listed above, may have caused the low/absent dissociative states measured at T2 after the mirror-gazing task. The authors should declare these errors in a sub-chapter within the General Discussion of their main manuscript. This sub-chapter should be entitled "Limitations" and placed toward the end of their revised paper. These declarations might exclude (or reduce) the likelihood of persevering into similar flawed executions of the mirror-gazing procedure and setup.

Response:

The requested section was added to the limitations paragraph in the General Discussion (lines 643-646). We wish to note, however, that there were significant changes in pre- to post-mirror-gazing dissociation (and not low/absent). Notably, Caputo [2] himself, when describing the mirror gazing paradigm, noted that “the details of the setting (in particular the room illumination) are not critical.” This remark influenced our decision to use the Caputo procedure as a guidance, or a source of inspiration, and not aim for an immaculate replication. We now underscore that our protocol was not identical to Caputo’s but rather introduced some adaptations (lines 104-110).

6. The face conveys one's identity. Hence, mirror-gazing certainly involves both face perception and identity recognition. Therefore, limiting the effect of prolonged mirror gazing only to "dissociative detachments" is incorrect. A famous paper on dissociative states (Holmes et al, 2005, Clinical Psychology Review, not cited) distinguished "dissociative detachments" and "compartmentalizations" of other identities. The authors should update and correct their incomplete theoretical approach to mirror-gazing.

Response:

We did not elaborate on the conceptualization of dissociative detachments and compartmentalizations [5], as we sensed that it is not an essential notion for this investigation. Originally, we had mentioned briefly (only in the abstract) that the observed phenomena in our experiments were dissociative detachments. Following this comment, however, we deleted this definition from our abstract, as the abstract should not contain information that is not discussed in the manuscript itself.

Still, we see fit to respond here regarding how we see Holmes’ et al.’s conceptualization in the context of our study. Detachments, according to Holmes, include separation of the self from certain aspects of everyday experience. Compartmentalizations, on the other hand, are characterized by difficulty controlling actions, or cognitive processes, which are usually voluntarily controlled. The most prominent aspect of compartmentalization is that the compartmentalized processes continue, although being inaccessible, and their intact flow affects emotion, thoughts and actions. In other words, the main difference between detachments and compartmentalizations according to Holmes et al. is the presence of “split” functions that operate independently. Holmes and colleagues emphasize that detachments and compartmentalizations are usually distinguished and do not occur together. As we understand their conceptualization, the mirror-gazing paradigm causes a detachment from a certain aspect of everyday life, specifically, the familiar appearance of oneself. Although the paradigm certainly influences aspects of identity experience, we do not recognize that it generates a compartmentalized process that continues to operate independently and autonomously. Therefore, we believe that the experience it evokes cannot be classified as compartmentalization.

7. The measure (CADSS) used by the authors for DPDR is inadequate when evaluating mirror-gazing dissociation since it contains no items concerning mirrors. At the same time, DES has only one item dedicated to mirror gazing, and this item could explain the effect of "trait dissociation" that the authors found with various statistical interactions, given that DES (i.e., measurement of trait dissociation) was administered "after" the dissociative task (line 307). In the MGS group, the "suggestion" of paying attention to the face probably did not introduce a bias, as the authors suppose. Instead, it may enhance awareness of face changes in participants who, as a consequence, scored higher on DES.

Response:

As we see it, to establish that the clinical construct termed “dissociation,” harboring the widely-accepted meanings and definitions of this label, is indeed the outcome induced by mirror-gazing, it should be measured using its validated and well-known general scales, without necessarily asking specifically about a reflection in the mirror. Using a general validated state assessment tool for dissociation supports the convergent validity of the measured construct and shows that the resulting phenomenon is broader than momentary perceptual distortions regarding reflection, encompassing the broader aspects of the dissociative experience.

Regarding the mirror item in the DES, we thank the reviewer for this important observation, whereby results we considered as “trait” could have been affected by state influences on the mirror item. Following this observation, we conducted again all analyses involving the DES while excluding the mirror item, which may have been affected by the manipulation. Results remained unchanged. This is now mentioned in the Results sections of Study 1 and Study 2 (lines 296-301, 475-475).

8. Supplementary materials:

The authors only mention early explanations of mirror-gazing "dissociative detachments" and omit more recent theoretical accounts. It should be noted that these "dissociative detachments" are specifically linked to face stimuli, thus leading to identity discontinuities since the face affords one's identity. Therefore, from the very beginning, these "dissociative detachments" of one's face can exclude the simple explanation based on the Troxler effect. All three dissociative states (i.e., derealization, depersonalization, and dissociative identity) may be involved in "dissociative detachments" [and compartmentalization of new strange identities]. Indeed, Caputo (2023, Journal Trauma Dissociation, not cited) adopted this theoretical perspective and found that the CADSS scale is not an optimal scale for measuring mirror-gazing "dissociative detachments" [and identity compartmentalization] (indeed, this CADSS inadequacy is confirmed by the description of unsettling experiences of mirror gazing by one participant: lines 290-296).

Response:

Thank you for the important clarification regarding the theoretical explanations of the strange face illusion included in the supplementary materials file. We have now added newer perspectives and the missing citations to the relevant section describing the scientific understanding of the mechanisms governing the effect.

Regarding the comment about measurement: since the current investigation does not focus on the strange-face illusion itself but rather on its psychological impact, and specifically, on the increase in dissociative states that follows it, we considered it important to use a dedicated measure of dissociation, such as the CADSS. However, it is certainly possible that we would have benefited from a dedicated measure of the illusion as well. Using the SFQ could have helped us better organize and understand what our participants experienced during the experiment. We

---

## [Decision Letter · Decision Letter 1]

26 Nov 2025

Dear Dr. Bregman-Hai,

Thank you for revising your submission. You will see that Reviewer 1 raises a few outstanding points. If you would please address these, amending the manuscript where appropriate, I will then be able to reconsider your manuscript for publication.

We look forward to receiving your revised manuscript.

Kind regards,

Clare Eddy

Academic Editor

PLOS ONE

Journal Requirements:

Reviewers' comments:

Reviewer's Responses to Questions

**Comments to the Author**

Reviewer #1: (No Response)

Reviewer #2: All comments have been addressed

2. Is the manuscript technically sound, and do the data support the conclusions?

Reviewer #1: Yes

Reviewer #2: Yes

3. Has the statistical analysis been performed appropriately and rigorously?

Reviewer #1: Yes

Reviewer #2: Yes

4. Have the authors made all data underlying the findings in their manuscript fully available?

Reviewer #1: Yes

Reviewer #2: Yes

5. Is the manuscript presented in an intelligible fashion and written in standard English?

Reviewer #1: Yes

Reviewer #2: Yes

Reviewer #1: The authors do not mention the main limitation of their work: the complete absence of controls during the participants' correct execution of the experimental task. This drawback is particularly critical in the mirror-gazing task because this task can produce unsettling experiences in some participants (see the typical experience of depersonalization/derealization that is reported in lines 325-331). In their response, the authors gave a weak justification for their neglect, given that the presence of a micro-camera across all experimental conditions produced a similar effect across all experimental conditions, offering the opportunity to monitor the participant without the need for an experimenter in the laboratory. Conversely, monitoring of correct gazing through a video-camera can be very effective in mirror-gazing tasks: see standard measures of dissociation in Caputo (2023, Journal Trauma Dissociation, not cited).

The authors should include this limitation in the final section of the discussion. It is important to acknowledge this limitation, as future studies (possibly by the authors themselves) should keep this in mind.

line 31. "dissociative absorption" - This is an incorrect definition that creates confusion: normal absorption is not a dysfunctional dissociative state. (Dalenberg, C. J., Katz, R. R., Thompson, K. J., & Paulson, K. (2022). The case for the study of “normal” dissociation processes. In: Martin J. Dorahy, Steven N. Gold, John A. O’Neil (eds.), Dissociation and the dissociative disorders (pp. 81-92), not cited). The authors should not insist in this error concerning dissociation that is still present in the literature.

lines 112-115. "As we read the Caputo [26] procedure, we noticed that participants were informed in advance that they might notice perceptual changes in their reflections. We speculated that such a warning might have influenced Caputo’s results as it contained a potentially suggestive cue." This is an outdated, ill-posed question to which the article mentioned (Caputo, 2023) has already responded. This ill-posed question concerning supposedly the effects of the "suggestion" is not correct. Indeed, these effects are simply caused by differences in working memory storage, an issue common to many cognitive and perceptual behaviors when the task for participants to perform is lacking, yielding no memory traces. It is not caused by the real existence of changes to the level of dissociation by the mirror-gazing task, but only by the absence of a task to encourage participants in the MG condition (compared to MGS condition). In fact, the level of dissociation can remain the same, although a difference is detected by self-report questionnaires of dissociation that are based on conscious memory.

Reviewer #2: (No Response)

**Do you want your identity to be public for this peer review?** For information about this choice, including consent withdrawal, please see our Privacy Policy

Reviewer #1: No

Reviewer #2: No

---

## [Author Response · Author response to Decision Letter 2]

8 Dec 2025

Response to Reviewers

I thank reviewer #1 for their comments. I have revised the manuscript accordingly, and my detailed responses are provided below.

Reviewer #1:

1. The authors do not mention the main limitation of their work: the complete absence of controls during the participants' correct execution of the experimental task. This drawback is particularly critical in the mirror-gazing task because this task can produce unsettling experiences in some participants (see the typical experience of depersonalization/derealization that is reported in lines 325-331). In their response, the authors gave a weak justification for their neglect, given that the presence of a micro-camera across all experimental conditions produced a similar effect across all experimental conditions, offering the opportunity to monitor the participant without the need for an experimenter in the laboratory. Conversely, monitoring of correct gazing through a video-camera can be very effective in mirror-gazing tasks: see standard measures of dissociation in Caputo (2023, Journal Trauma Dissociation, not cited).

The authors should include this limitation in the final section of the discussion. It is important to acknowledge this limitation, as future studies (possibly by the authors themselves) should keep this in mind.

Response:

The requested limitation was added to the limitations paragraph (lines 658-661).

2. Line 31. "dissociative absorption" - This is an incorrect definition that creates confusion: normal absorption is not a dysfunctional dissociative state. (Dalenberg, C. J., Katz, R. R., Thompson, K. J., & Paulson, K. (2022). The case for the study of “normal” dissociation processes. In: Martin J. Dorahy, Steven N. Gold, John A. O’Neil (eds.), Dissociation and the dissociative disorders (pp. 81-92), not cited). The authors should not insist in this error concerning dissociation that is still present in the literature.

Response:

We believe that our approach completely aligns with the chapter referenced in this comment, as the authors of that chapter conclude that absorption is part of the dissociative domain. For example, in their concluding remarks, their first conclusion is: "Empirical evidence, the evidence that should ultimately guide us, suggests that "normal" dissociation, or absorption, is a form of dissociation". (pp. 89 in Dalenberg et al., The case for the study of "normal" dissociation processes, In: Dorahy et al., 2022, Dissociation and the Dissociative Disorders). However, as the term "dissociative absorption" bothered the reviewer, we changed it. In the previous version of the manuscript, it was only used once - in the abstract ("...but the control condition was not sufficiently differentiated, also increasing dissociative absorption"). In the new revised version of the manuscript, we changed to the following:

"...but the control condition was not sufficiently differentiated, as it also increased absorption, a construct which is considered as part of the domain of dissociative experiences".

3. Lines 112-115. "As we read the Caputo [26] procedure, we noticed that participants were informed in advance that they might notice perceptual changes in their reflections. We speculated that such a warning might have influenced Caputo’s results as it contained a potentially suggestive cue." This is an outdated, ill-posed question to which the article mentioned (Caputo, 2023) has already responded. This ill-posed question concerning supposedly the effects of the "suggestion" is not correct. Indeed, these effects are simply caused by differences in working memory storage, an issue common to many cognitive and perceptual behaviors when the task for participants to perform is lacking, yielding no memory traces. It is not caused by the real existence of changes to the level of dissociation by the mirror-gazing task, but only by the absence of a task to encourage participants in the MG condition (compared to MGS condition). In fact, the level of dissociation can remain the same, although a difference is detected by self-report questionnaires of dissociation that are based on conscious memory.

Response:

Following this comment, we have refined our approach to the subject of suggestion. First, because our data collection was conducted before the publication of the 2023 research, we were not familiar with that article when designing the present study. Second, we believe that our findings contribute to existing knowledge, as we directly compared, within the same research design and sample, participants who received suggestive instructions with those who received more general instructions. Furthermore, our findings align with the 2023 results. We now note this both toward the end of the Introduction section (lines 120-129) and in the Discussion of Study 2 (lines 569-571 ).

---

## [Decision Letter · Decision Letter 2]

6 Jan 2026

Mirror-gazing-induced dissociation impairs self-reported and implicit sense of agency: A causal investigation of dissociation and agency under controlled laboratory conditions

PONE-D-25-34409R2

Dear Dr. Bregman-Hai,

We’re pleased to inform you that your manuscript has been judged scientifically suitable for publication and will be formally accepted for publication once it meets all outstanding technical requirements.

Kind regards,

Clare Eddy

Academic Editor

PLOS One

Additional Editor Comments (optional):

Reviewers' comments:

Reviewer's Responses to Questions

**Comments to the Author**

Reviewer #1: All comments have been addressed

2. Is the manuscript technically sound, and do the data support the conclusions?

Reviewer #1: Yes

3. Has the statistical analysis been performed appropriately and rigorously?

Reviewer #1: Yes

4. Have the authors made all data underlying the findings in their manuscript fully available?

Reviewer #1: Yes

5. Is the manuscript presented in an intelligible fashion and written in standard English?

Reviewer #1: Yes

Reviewer #1: (No Response)

**Do you want your identity to be public for this peer review?** For information about this choice, including consent withdrawal, please see our Privacy Policy

Reviewer #1: No

---

## [Editor Report · Acceptance letter]

PONE-D-25-34409R2

PLOS One

Dear Dr. Bregman-Hai,

I'm pleased to inform you that your manuscript has been deemed suitable for publication in PLOS One. Congratulations! Your manuscript is now being handed over to our production team.

Kind regards,

on behalf of

Dr. Clare Eddy

Academic Editor

PLOS One